# ReMix: Optimizing Data Mixtures
# for Large Scale Imitation Learning

**Joey Hejna**
Stanford

**Chethan Bhateja**
Stanford

**Yichen Jiang**
Stanford

**Karl Pertsch**
Stanford, UC Berkeley

**Dorsa Sadigh**
Stanford

**Abstract:** Increasingly large imitation learning datasets are being collected with the goal of training foundation models for robotics. However, despite the fact that data selection has been of utmost importance in vision and natural language processing, little work in robotics has questioned what data such models should actually be trained on. In this work we investigate how to weigh different subsets or "domains" of robotics datasets for robot foundation model pre-training. Concretely, we use distributionally robust optimization (DRO) to maximize worst-case performance across all possible downstream domains. Our method, Re-Mix, addresses the wide range of challenges that arise when applying DRO to robotics datasets including variability in action spaces and dynamics across datasets. Re-Mix employs early stopping, action normalization, and discretization to counteract these issues. Through extensive experimentation on the largest open-source robot manipulation dataset, the Open X-Embodiment dataset, we demonstrate that data curation can have an outsized impact on downstream performance. Specifically, domain weights learned by Re-Mix outperform uniform weights by 38% on average and outperform human-selected weights by 32% on datasets used to train existing generalist robot policies, specifically the RT-X models.

**Keywords:** Data Curation, Data Quality, Robot Imitation Learning

## 1  Introduction

Many breakthroughs in machine learning can be attributed to "Internet-scale" datasets, from the development of vision models like CLIP [1] to advancements in transformer-based language modeling powered by the Common Crawl dataset [2]. Seeking to capitalize on this trend, several recent efforts in robotics focus on collecting [3–6] or pooling [7] large scale robotics datasets with the goal of training more performant imitation learning policies. Learning from this data, however, is particularly challenging: robotics datasets are collected with different robots, environments, state spaces, action spaces, and dynamics [8]. For example, the commonly used Bridge V2 Dataset [4] uses a third person camera on a small WidowX robot and cartesian delta control, while many datasets [9–12] collected on the popular and much larger Franka Panda robot use wrist cameras [3] or joint-space actions [13]. While embracing such heterogeneity quickly scales the amount of available training data [7], it amplifies the importance of a fundamental question: how do we curate these raw, heterogeneous data sources into effective training datasets for generalist robot policies?

While early vision and language models were trained on highly-curated academic datasets such as ImageNet [14], questions surrounding data selection have shaped modern training pipelines that use Internet-scale data [15–17]. For example, training large language models involves numerous stages of data filtering [18]. Similarly, large vision datasets, e.g. LAION [19], use pre-trained models to assess data quality. Thus as we scale robot data, curation weill become critical. Unfortunately, prior filtering techniques are often inadequate in robotics; we cannot use n-grams and visual embeddings do not capture the sequential nature of episodic data.

Even though aspects of demonstration data such as action quality [20] and visual diversity [3, 4, 21] have been shown to be of paramount importance to downstream performance, approaches for robotics data curation remain limited. In imitation learning (IL), the data selection problem has only been characterized theoretically [22, 23] or in simple small-scale settings [24]. Thus in practice we are left with ad hoc

---

[1] Correspondence to jhejna@cs.stanford.edu. Code: https://github.com/jhejna/remix

8th Conference on Robot Learning (CoRL 2024), Munich, Germany.

solutions. For example, though the Open-X-Embodiment dataset (OpenX) [7] is comprised of more than 60 individual datasets totalling over 2M robot trajectories, the RT-X models released alongside it were trained on a mixture of only 12 datasets, weighted based on expert intuition. The recently released Octo [25] and OpenVLA [26] generalist policies were similarly trained on a subset of OpenX, where the authors chose which datasets to include at what sampling weight based on a subjective notion of "interestingness". While the resulting data mixes are shown to work well in practice, their curation requires extensive domain knowledge and manual data inspection. Such ad hoc selection strategies are unlikely to scale to the rapidly growing datasets used to train robot policies [3, 5, 27].

In this work, we ask: how can we *automatically* curate large-scale robotics datasets to maximize the performance of generalist IL policies across domains? Though many filtering techniques are not directly applicable to robotics, we can borrow ideas from language modeling that systematically optimize data mixtures based on model performance. Specifically, DoReMi [28] uses group distributionally robust optimization [29] to maximize performance across "domains" in a given dataset. In the context of robotics, such "domains" can correspond to different scenes within a single dataset, e.g., different toy kitchens in Bridge V2 dataset [4], or can refer to full robot datasets in mixtures such as the OpenX dataset. However, due to the heterogeneity of robotics datasets we find that naively applying such techniques does not work. Distributionally robust optimization approaches minimize worst-case loss. Differences in action spaces and their distributions can cause loss magnitudes to be imbalanced across domains, leading some domains to be weighed more heavily than they should be. Moreover, the smaller size of robotics datasets makes overfitting easy. Both of these issues result in poor estimates of model performance, and consequently bad mixture weights.

To address these problems, we propose Re-weighing Robotic Dataset Mixtures with Minimax Optimization (Re-Mix for short), which instantiates the data curation problem as a min-max optimization, where a policy *minimizes* its excess behavior cloning loss over a reference model subject to learned domain mixture weights that try to *maximize* it. Intuitively, the excess loss measures how much room the policy has to improve on a given domain, and the data mixture is optimized to maximize such improvement potential. Crucially, we carefully control the loss magnitudes between domains via domain-independent action normalization and discretization, even if the final policies we train are continuous diffusion models [30, 31]. Moreover, we find that selecting a reference model that has not overfit to any domain prevents drastic skewing of the downstream domain weights.

We empirically evaluate Re-Mix by using it to automatically optimize the training data mixture for the Bridge V2 dataset [4] and the OpenX-based dataset used to train RT-X [7]. We show that policies trained with our data mix improve performance by 38% and 32% respectively over naïve data balancing and human-expert-curated data mixtures in evaluations using WidowX and Franka robot arms. Additionally, we show that weights from Re-Mix can effectively *sub-sample* both datasets, achieving competitive performance when using only 25% of the original data, while using uniform or human curated weights significantly reduces performance. Our contributions are as follows:

- We introduce Re-Mix, a method for automatically curating large-scale robotics datasets using group distributionally robust optimization over the behavior cloning loss.
- We demonstrate Re-Mix's ability to curate effective training data mixtures for the Bridge [4] datasets and the subset of the OpenX dataset [7] used to train the RT-X models.
- We curate 25% subsets of the Bridge and OpenX datasets which can be used for training generalist policies with minimal loss in performance, while reducing the required compute budget.

## 2  Related Work

In congruence with the rise of deep learning in various fields, data selection has become of increasing interest. Here we review the most relevant works, organized by area.

**The Data Problem in Robotics.** Several recent works in robotics have focused on collecting large demonstration datasets for imitation learning in simulation [20, 32, 33] and the real world [3, 7, 34–38] to train large-scale robot policies [6, 25, 39, 40]. Generally, these works along with others that study the influence of data collection on compositional generalization [21, 41, 42] show that aspects of dataset construction such as scene and task diversity have a direct impact on downstream policy generalization. Though several studies focus on *how* data should be collected via specific hardware [43], collection procedures [11, 21, 44],

or provide theoretic insights about data collection [22], little work in robotics addresses the post-hoc dataset selection and analysis problem. This is particularly important as the number and diversity of robot datasets are increasing with less clear conclusions about how to train a policy that effectively consumes all the collected data [3, 7, 25]. Baker et al. [45] train a classifier to predict data quality, but require human annotations which are impractical to scale. Perhaps most related are retrieval-based methods that subset datasets [12, 46], but do so based on a priori target task specifications and are thus inapplicable to training generalist policies.

**Data Curation in Computer Vision.** Computer vision datasets were originally hand-crafted and manually labeled [14, 47]. However, scaling datasets to beyond what is possible to curate by hand, while retaining quality, has been critical to increasing performance [1, 48]. Notably, filtering techniques based on metadata-count balancing [49], embeddings [19], optical flow [50], and clustering [51] have shown to greatly improve downstream performance despite filtering out large amounts of data. At the extreme, coreset selection methods use active learning [52, 53], but have prohibitive computational requirements [54, 55] Data curation techniques from computer vision can only filter state-action trajectories in an action-agnostic manner – potentially removing useful parts of a dataset.

**Data Curation in Natural Language Processing.** When training on large real-world sources of text, language modeling pipelines employ a number of text-specific preprocessing steps including metadata, dialect, de-duplication, and toxicity filtering [16, 18, 56, 57]. More advanced methods also consider sub-setting data to maximize downstream performance, as in this work, but use techniques such as k-means clustering over embedded text [58, 59]. While such clustering techniques are potentially visually informative in robotics – similar to curation works in computer vision – they do not provide information about *actions*. Mixture techniques, such as Domain Reweighting with Minimax Optimization (DoReMi) [28] balance text domains using robust optimization and build upon ideas from prioritized training [60–62]. Our work is inspired by DoReMi as such robust optimization approaches can be applied to imitation learning as well. In this work, we discuss the challenges of applying these techniques in robotics, and propose a solution that addresses their limitations for effective dataset curation for imitation learning.

## 3 Re-weighing Robotic Dataset Mixtures with Minimax Optimization

In this section, we first formalize the problem of re-weighting data mixtures for IL. We then discuss our approach which uses distributionally robust optimization to select domain weights.

**Problem Setup.** We consider the general imitation learning problem, where we are given a dataset of demonstrations $\mathcal{D} = \{\tau_1, ..., \tau_n\}$ consisting of state-action trajectories $\tau = (s_1, a_1, ..., s_{T_i}, a_{T_i})$. Our goal is to learn a parameterized policy $\pi_\theta$ that learns a mapping from states to actions $\pi_\theta : \mathcal{S} \to \mathcal{A}$. In practice, this is often done through standard imitation learning algorithms such as behavior cloning (BC) by minimizing the expected negative log-likelihood of the actions under the policy:

$$\mathcal{L}_{\text{BC}}(\pi_\theta, \mathcal{D}) = \mathbb{E}_{(s,a) \sim \mathcal{D}}[-\log \pi_\theta(a|s)] \tag{1}$$

However, datasets often contain more information than just state action pairs. We assume that the overall dataset $\mathcal{D}$ can be split into $k$ heterogeneous domains $\mathcal{D}_1, ..., \mathcal{D}_k$. This is a general assumption: while these domains could be larger groups, like different datasets from the Open X-Embodiment dataset [7] with different embodiments, they could also be as small as single trajectories. Moreover, each of the $k$ domains can differ in state space $\mathcal{S}$, action space $\mathcal{A}$, transition dynamics, or their distributions. In fact when learning large behavior models, such heterogeneity becomes necessary to access more sources of diverse data. In this work, we use the Bridge dataset [4] – with different environments as the domains, and the Open-X-Embodiment dataset [7] – with different robot embodiments as the differing domains.

Our goal is to learn a weighting vector $\alpha \in \Delta^k$ that specifies a probability distribution over all domains such that any model, when trained on a domain mixture weighted according to $\alpha$, attains maximum performance *across all domains*. Unlike the data retrieval, where data is selected *for a particular target task*, our goal is to curate datasets for effective pre- or co-training without any knowledge of a target task.

**Distributionally Robust Optimization.** When pre-training on large amounts of robot data we want policies to *generalize* to new settings and tasks, not master a specific target task. With that in mind, we want to optimize for a data mixture that results in models that i) can perform as well as possible on

each domain, but ii) do not overfit to any one domain at the expense of another. Distributionally robust optimization (DRO) techniques aim to solve the same problem: learn models that minimize the worst-case training loss [29] – BC loss in the case of imitation learning – across domains $\mathcal{D}_1 ... \mathcal{D}_k$. Specifically, naïvely applying group robust optimization techniques in robotics would result in the following objective:

$$\min_\theta \max_{\alpha \in \Delta^k} \sum_{i=1}^k \alpha_i \mathcal{L}_{BC}(\pi_\theta, \mathcal{D}_i). \tag{2}$$

With this objective, $\alpha$ up-weights domains that have a higher loss value, emphasizing the hardest domains. However, in practice we might not be interested in just fitting the domains with higher losses. For example, a dataset with complex rotations may always have higher BC loss than simple pick-place datasets. Thus, standard robust optimization techniques could end up ignoring the latter domain. Instead, as in prior work [28, 52, 63], we consider the *difference* in loss between our learned policy $\pi_\theta$ and a reference policy $\pi_{ref}$ which is trained to convergence on an initial guess of the domain weights, usually assumed to be proportional to the size of each domain, i.e. uniform sampling. In Eq. (2) this equates to replacing $\mathcal{L}_{BC}$ with $\mathcal{L}_{BC}(\pi_\theta, \mathcal{D}_i) - \mathcal{L}_{BC}(\pi_{ref}, \mathcal{D}_i)$. We refer to this difference as the *excess loss*, and use it for robust optimization. Like before, this will down-weight domains that the policy fits well, as it can achieve a loss similar to that of the reference model. However, it crucially also down-weights domains which are difficult to fit (i.e. they have a high policy loss *and* a high reference loss) due to the relative nature of the excess loss. Therefore, only domains that have a high excess loss, meaning the policy can improve to match the reference model, will be up-weighted as $\alpha$ is chosen to maximize the excess overall loss.

Unfortunately, models trained directly with robust optimization often exhibit worse overall performance [64, 65]. Alternatively, we can use the learned $\alpha$ for downstream training as in Xie et al. [28]. This gives us a set of reusable weights that can be used to train different policies without the need for robust optimization.

### 3.1 The Challenges of Applying Robust Optimization in Robotics

While Group DRO has been applied in language modeling [28], robust optimization techniques face unique challenges in robotics which we highlight here. We then detail how we adapt a distributionally robust optimization pipeline to select domain weights for robotics datasets.

**Unbalanced Losses.** Large robotics datasets are often highly hetero-geneous: many are collected across different embodiments, controllers, frequencies and even units (e.g., inches vs meters). Within the same dataset, different scenes or tasks require vastly different ranges and speeds of motion. As a result, some datasets may have an outsized effect on

|  | $\alpha_{noise}$ | $\alpha_{bridge}$ |
|---|---|---|
| Bounds | 0.943 | 0.057 |
| Gaussian | 0.158 | 0.842 |

**Table 1:** Learned $\alpha$ from toy setting in Section 3.1

robust optimization. To address this issue, one needs to align action losses across domains. Action normalization is often applied in imitation learning to standardize datasets to a common distribution [20, 25]. In our case, we specifically apply Gaussian normalization to each embodiment *individually*. We note that bounds normalization [30] applied to each domain, would be insufficient as it would not align the moments of each domains action distribution. To underline the importance of aligning actions to a common distribution, we construct a simple experiment by training a policy with Group DRO [29] (Eq. (2)) when the action distributions match versus when they differ. Specifically, we construct a *noise* domain where a subset of the Bridge V2 dataset [4] is assigned random Gaussian actions and a *bridge* domain which uses the original actions, either normalized to also be unit gaussian or rescaled between -1 and 1 using "bounds" normalization. When Gaussian normalization is applied to the *bridge* domain, the action distribution matches the random noise unlike with bounds normalization. We show the learned domain weights $\alpha$ for each scheme in Table 1. While one might expect that $\alpha$ assigns majority weight to the *bridge* domain, as the *noise* domain is impossible to fit, this is actually only true in the "Gaussian" case when the action distributions of are aligned.

**Continuous Losses.** Robust optimization has largely been applied in discrete classification problems with cross-entropy losses, for example in language modeling [63]. Popular policy learning approaches, however, often predict continuous actions and use L1 or L2 loss functions [20, 30, 66, 67]. Applying robust optimization in these settings can be problematic for two reasons. First, action distributions can be multi-modal, and expressive continuous policy classes such as diffusion models only optimize an upper bound on the true loss. However, without the expressiveness to fit multi-modal distributions, both the reference policy and DRO would be unable to effectively minimize BC loss on domains with multi-modal actions. Second,

compared to language datasets, robot datasets often have a large number of action outliers which can heavily sway the value of continuous action losses. With L1 or L2 loss, these outliers can significantly increase the loss of a given domain, causing DRO to believe it can still make progress. To resolve these problems, when applying robust optimization in the robotics domain, we discretize each action dimension via binning.

**Overfitting.** Datasets in language modeling often contain billions of tokens. As a result, robust optimization techniques like Xie et al. [28] do not experience overfitting when applied to these large scale datasets. On the other hand, large robot datasets are comparatively small ($\sim$10-100k demonstrations). In this regime, it is common for high-capacity policies to achieve near-zero training loss for every datapoint [6, 25, 26, 68]. This is problematic when using the excess loss for robust optimization: if the reference model achieves near-zero training loss on every data point within a domain, the excess loss is equivalent to the regular loss (since the reference loss is always $\simeq 0$) and $\alpha$ no longer reflects the potential for improvement on each domain. To counteract this problem, we employ aggressive early stopping on both the reference model and robust optimization. Specifically, we select the latest checkpoint from the reference model that has not overfit to *any* of the domains $\mathcal{D}_1,...,\mathcal{D}_k$ as measured by the difference in loss values between the training dataset and a held-out validation dataset for the respective domain. This ensures that the reference model does not overfit to any individual domain and the learned weights $\alpha$ are informative.

### 3.2  Re-weighing Robotic Dataset Mixtures with Minimax Optimization

Our approach, Re-Mix, uses group distributional robustness to determine the weights of a data mixture [28] that could then be used for policy training and incorporates the key design considerations from the previous section, addressing issues around unbalanced losses, continuous losses, and overfitting. We note that Re-Mix only returns the weights of the data mixture $\alpha$, as opposed to the final policy. This is to decouple the data curation problem from the policy training problem. After running Re-Mix, the resulting weights can be used for learning policies of a different type (i.e. diffusion) or at a larger scale.

**Stage 1: Action Preprocessing.** Following Section 3.1, we apply Gaussian normalization separately to every domain $D_k$ with different action spaces and dynamics, and then discretize actions via binning.

**Stage 2: Reference Model Training.** Next, we train a discrete reference model $\pi_{\text{ref}}$ on the uniform mixture of domains $\mathcal{D}_1,...,\mathcal{D}_k$, where each domain is weighted in proportion to its size. We select the final reference model checkpoint by validation loss per Section 3.1, and use it to estimate the excess loss per domain.

**Stage 3: Group Distributionally Robust Optimization.** We learn the domain weights $\alpha$ via the following robust optimization with a discrete policy $\pi_\theta$:

$$\min_\theta \max_{\alpha \in \Delta^k} \sum_{i=1}^k \alpha_i \left[ \frac{1}{|\mathcal{D}_i|} \sum_{(s,a) \in \mathcal{D}_i} (-\log \pi_\theta(a|s) + \log \pi_{\text{ref}}(a|s)) \right], \tag{3}$$

which minimizes the worst case excess BC loss of the learned policy $-\log \pi_\theta(a|s) + \log \pi_{\text{ref}}(a|s)$ over all possible weightings of the domains $\alpha \in \Delta^k$. To update $\alpha$, following [29], we perform one step of exponentiated gradient ascent on $\alpha$ followed by domain-weighted gradient descent on $\theta$ at each training step. Our resulting values of $\alpha$ upweight domains that we can still improve on, while downweighting domains that are trivial or impossible to fit. This means that Re-Mix directly filters data based on actions, unlike other techniques in vision and language that solely filter based on embeddings [59, 69]. We optimized Eq. (3) for the same number of steps as the reference model.

**Stage 4: Data Weighting for Policy Training.** After robust optimization over the excess loss, we take the average value of $\alpha$ over the course of training, which we denote by $\bar{\alpha}$. We can then use this value of $\bar{\alpha}$ to re-weight different domains, or even subset datasets for policy training. In practice, this means that we can re-use the weights for several training runs with different configurations. For example, Re-Mix uses discrete actions, but we train final policies with diffusion which has shown to perform well empirically [25, 30].

## 4  Experiments

We aim to answer the following questions: (1) Does Re-Mix effectively curate large robot datasets for downstream policy learning? (2) Can we use Re-Mix to heavily sub-sample robot datasets while retaining good performance? (3) Which design decisions matter for effective automatic curation of large robot datasets?

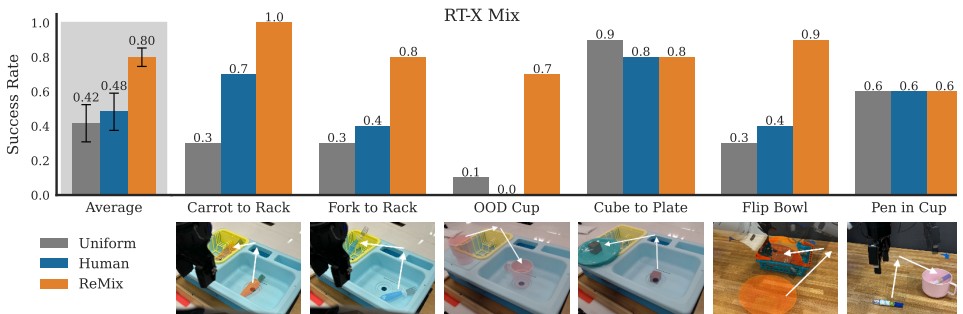

**Figure 2: Results for curating the RT-X training mix.** We test policies trained on different weightings of the data mixture used by RT-X across two WidowX (left) and two Franka (right) tabletop manipulation tasks. We find that the policy trained on the data mix curated with Re-Mix achieves strongest performance, even outperforming the human-expert-curated data mix from RT-X [7]. Mean ± StdErr across 4 tasks, 10 evaluations each.

## 4.1 Experimental Setup

**Datasets.** We test Re-Mix curation on two widely-used, large-scale robot datasets: (1) the Bridge V2 Dataset [4], consisting of 50k diverse teleoperated demonstrations of single-arm manipulation tasks with a WidowX 6 DoF robot arm, and (2) the datasets from the Open X-Embodiment dataset used to train RT-1-X and RT-2-X models [7] which have third-person cameras, consisting of a total of 350k demonstrations which span disparate embodiments and environments. We use "RT-X" to refer to this set of datasets. We partition the Bridge V2 dataset into 32 domains based on the scenes the data was collected in. For OpenX, we use each of the 11 datasets in the RT-X training set as a domain for our curation experiments. The OpenX data mix is particularly challenging for effective curation due to its heterogeneous data sources. For a detailed list of all datasets and partitions, see Appendix B. For simulation experiments see Appendix A.

**Training and Evaluation Details.** We aim to assess the quality of various curated pre-training data mixtures for downstream policy learning. To that end, we co-train generalist goal-conditioned policies on the curated datasets. As we do not have access to the robot setups used to collect the datasets we train on, we construct our own WidowX and Franka robot evaluation setups. Unfortunately, policies trained on *only* the pre-training data failed to zero-shot generalize to our out-of-distribution setups. To address this, we follow prior works [3, 12, 46, 70] and co-train our policies on a small amount of in-domain data (25 demonstrations each for 3 representative tasks), added to the final training mixture at a small weight of 5%. We then evaluate policies on tasks that are out-of-distribution with respect to the co-training data to test generalization. As a result of

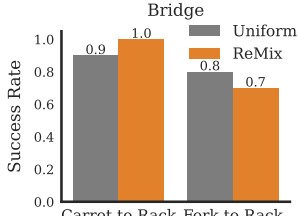

**Figure 1:** On Bridge V2 [4] there is no notable difference between uniform sampling vs. Re-Mix when training on the full dataset.

co-training, all policies achieve non-zero success rate. However, we note that the in-domain dataset is small enough that the quality of the pre-training data mix still has significant impact on the evaluation result, providing a good test bed for data curation approaches. All models are evaluated in the real world with 10 trials per task totaling over 500 real-world trials cumulatively. For all policies we use a ResNet 50 image encoder [71]. For the Re-Mix reference model and Group DRO optimization, we use a discrete MLP action head. For all final policies we use the diffusion head from [4, 25, 72] and train all models for 400,000 gradient steps.

**Comparisons.** We compare the quality of Re-Mix's curated data mixes to a naïve baseline: sampling uniformly from each domain according to the total number of state-action pairs (**Uniform**). For evaluations on the OpenX datasets, we additionally compare to a human-expert-curated data mix, using the hand-crafted weights from RT-X [7]. For Bridge there is no expert-curated data mix — uniform sampling is the norm.

## 4.2 How do Re-Mix weights impact performance?

In Fig. 2, we show results for weighing datasets from the RT-X mix according to different methods. For the WidowX robot, we consider four tasks that test generalization to 1) unseen objects: "Carrot to Rack", "OOD Cup", 2) unseen initial conditions: "Fork to Rack", and 3) distractors at the goal location "Cube to

| Method | $\alpha_{\text{UR5}}$ | $\alpha_{\text{Cable Routing}}$ | $\alpha_{\text{Bridge}}$ | $\alpha_{\text{Jaco}}$ | $\alpha_{\text{Kuka}}$ | $\alpha_{\text{RoboTurk}}$ | $\alpha_{\text{RT1}}$ | $\alpha_{\text{Taco Play}}$ | $\alpha_{\text{Taco Extra}}$ | $\alpha_{\text{Toto}}$ | $\alpha_{\text{Viola}}$ |
|---|---|---|---|---|---|---|---|---|---|---|---|
| Uniform | 1.01% | 0.43% | 22.7% | 0.81% | 24.9% | 1.94% | 40.9% | 0.60% | 2.46% | 3.42% | 0.80% |
| Human | 1.22% | 1.56% | 27.5% | 1.95% | 25.1% | 2.35% | 26.8% | 1.46% | 5.94% | 4.13% | 1.90% |
| Re-Mix | 2.37% | 0.20% | 19.9% | 0.39% | 12.1% | 1.14% | 42.5% | 0.63% | 3.04% | 16.3% | 1.51% |

**Table 2:** Dataset mixture weights by different methods on the RT-X dataset mix [4, 6, 9, 10, 37, 73–76]. We color relative increases of more than 25% from uniform green and relative decreases of more than 25% red.

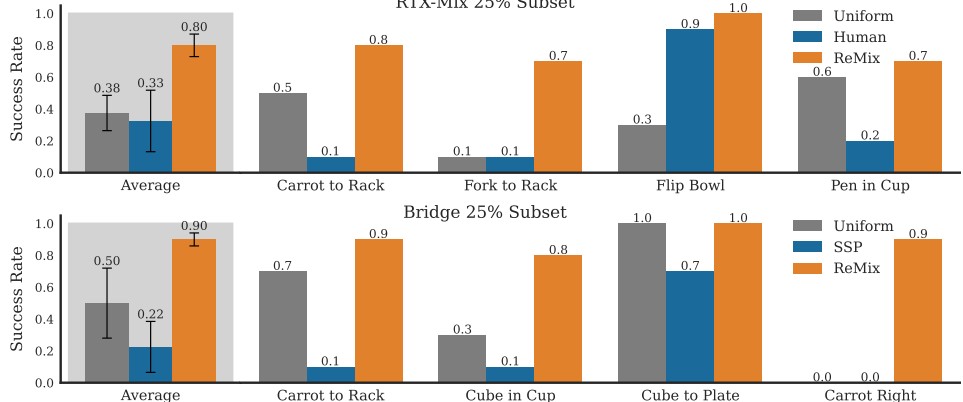

**Figure 3:** Results sub-setting datasets via different strategies until they reach 25% of their original size. We again use 10 evaluations per task, and show the Mean $\pm$ StdErr.

Plate". Similarly, for the Franka Panda robot we consider two tasks that test generalization to 1) unseen initial conditions "Pen in Cup" and 2) motions not seen in the RT-X data "Flip Bowl". Additionally, our Panda robot uses a Robotiq 2F-85 gripper, which was not present in any of the RT-X-datasets. Note that for the RT-X mix, we co-train the same model on both the WidowX and Franka data. As expected, we find that the domain weights selected by human experts for the RT-X models outperform the naïve uniform sampling baseline by 6% on average. More interestingly, we find that weighting datasets according to Re-Mix outperforms uniform weighting by 38% on average, and surprisingly outperforms the human curated weights by 32% on average. We find that Re-Mix generalizes better to unseen objects and locations in Bridge. Re-Mix performs particularly performs well for the "Flip Bowl" tasks, which is potentially because it up-weights the relatively small "Toto" dataset in OpenX that contains similar pouring motions to the flip blow task, though performance in Pen-in-cup is similar across all models. We hypothesize this is because it requires less generalization than the other tasks, and thus depends less on the pre-training mixes. Fig. 1 shows results using Re-Mix weights versus uniform weighting over scenes in the Bridge dataset. We find that performance in this setting is similar across both models. We posit that in the presence of the full Bridge dataset, selecting weightings is less important as the model is able to fit every scene well.

### 4.3 Analyzing Re-Mix Weights

Table 2 shows the weights produced by different methods on the RT-X dataset mix in comparison to the uniform mixture, which corresponds to sampling each datapoint with equal probability or equivalently weighting each domain by its total size (as fraction of the total number of datapoints). The human-expert-designed weights largely down-weight RT-1 [6], while up-weighting some of the smaller datasets like Routing [73], and Taco [9], perhaps to ensure they were sampled often enough to not be ignored. On the other hand, Re-Mix largely down-weights the Kuka dataset [76]. This dataset was autonomously collected and then filtered by success, making it of potentially lower action quality. Re-Mix also down-weights some smaller domains that are easy to fit; for example, Cable Routing has no gripper actions and Jaco [74] only has three possible actions. Surprisingly, Re-Mix up-weights the Toto dataset [77] by more than 4x. We posit that this is because Toto has a particularly multi-modal action distribution which deviates far from a standard Gaussian even after normalization and thus may be more challenging to fit. See Appendix A for a plot of its action distribution.

### 4.4 How well does Re-Mix subset datasets?

Though co-training on diverse data is important for performance [3, 70], doing so is often expensive given that modern robot datasets like the OpenX dataset encompass TBs of data. In this section, we evaluate how well Re-Mix can be used to *subset* datasets. The key idea: if Re-Mix weights are proportional to the importance of the data in each domain, we can use them to effectively sub-set the dataset by removing

data from domains that Re-Mix assigns low weight. We subset the base datasets according to Re-Mix and baselines by first computing the target size of the entire data mix *after* sub-setting, in our case 25% of $|\mathcal{D}|$. Then, we remove datapoints according to the mixture weights $\bar{\alpha}$. If a small dataset is upweighted too much (i.e. if domain $i$ is 1% of $|\mathcal{D}|$ but is upweighted to $\bar{\alpha}_i = 5\%$), there might not be enough data to exactly match $\bar{\alpha}$ from subsetting alone. Thus, after subsetting we sample the remaining points uniformly until we reach the target size. During training we still weight datasets according to the exact $\bar{\alpha}$.

The goal of subsetting is to retain performance when data is removed. We compare performance of subsetting with the Re-Mix weights to using the naïve uniform and human expert weights. For Bridge, where no expert weighting exists, we additionally compare to a vision subsetting method called "Self-Supervised Prototypes" (SSP) [69] which runs k-means on image embeddings and discards data closest to each centroid to encourage diversity. We average CLIP embeddings across each trajectory and use $k = 32$, matching the number of domains used by Re-Mix. To provide a more extensive evaluation on Bridge, we add two additional tasks. See Appendix B for details.

Subsetting results are shown in Fig. 3. Overall, we find that subsetting exacerbates the difference between methods, as the weights directly affect dataset composition. On the RT-X datasets (Fig. 3 top row) with only 25% of the data Re-Mix retains performance while human weights drop over 10%. This is likely because as shown in Table 2, Re-Mix places higher weights on some of the smaller datasets and down-weights some of the larger datasets such as the Kuka dataset from [40]. For example, when using domain weights for subsetting Re-Mix retains 72% of the UR5 Dataset but only 12% of the Kuka dataset, while the expert human weights retain only 30% of the UR5 Dataset but 24% of Kuka which is from suboptimal learned policies. Generally, the human weights do not down-weight datasets that are potentially uninteresting or not useful for training. On Bridge (Fig. 3 bottom row), Re-Mix also outperforms baseline methods. Overall SSP performs poorly, likely since robot trajectories may be out-of-distribution for vision models such as CLIP.

### 4.5 What matters in Re-Mix?

In this section, we ablate several design choices used in Re-Mix (see Section 3.1), including action discretization and early stopping. We run all ablations in the 25% subset setting (see Section 4.4), since subsetting further amplifies the effects of the domain weights. In Fig. 4, we first analyze the effects of choosing a reference model checkpoint for Group DRO that is overfit to the training dataset. Empirically, we find that choosing a checkpoint just 50K steps after early stopping decreases performance by over 15% on average, likely because the reference model baseline used to determine the domain weights is less meaningful once it overfits. On the right half of Fig. 4, we show performance on Bridge when using continuous (Cont.) actions in Re-Mix instead of discrete for estimating $\alpha$. We find that continuous actions lead to significantly worse performance, as their loss functions fail to fit outliers or multi-modal actions.

## 5 Limitations and Future Work

In this work we present Re-Mix, a method for automatically curating robotics datasets using distributionally robust optimization.

**Evaluation.** While we train on large, diverse robot datasets, the need for real world trials makes it difficult to exhaustively evaluate trained generalist policies on many robot embodiments and setups. While our evaluations capture two widely used robot arms from prior works [4, 7, 25], WidowX and Franka, future work should extend to more embodiments, perhaps via simulated environments [68].

**Abnormal Action Distributions.** We have noticed that Re-Mix upweights datasets with abnormal action distributions such as the Toto dataset. While resulting data mixes performed well, such up-weighting is not necessarily desirable. We hope to achieve less sensitivity to such irregularities in future work.

**Computational Cost.** Using our pre-computed weights can significantly reduce the compute required to train generalist policies. However, our approach for computing Re-Mix weights requires training policies on the full data twice, once for the reference model and once for Group DRO optimization. Future work can instead strive to curate datasets "on-the-fly" within one run.

**Scaling Up.** While we have demonstrated improvements on two large datasets, Bridge V2 and RT-X, scaling up to even larger ones such as the entire OpenX dataset [7] (>2M episodes) is an exciting extension.

**Acknowledgments**

Compute for this research was provided by a Google TPU Research Cloud Grant. This work was supported by NSF #1941722, ONR project #N00014-22-1-2293, DARPA grant #W911NF2210214, Stanford HAI, and TRI. JH is supported by an NDSEG Fellowship.

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

# A    Additional Results

## A.1    Ablations

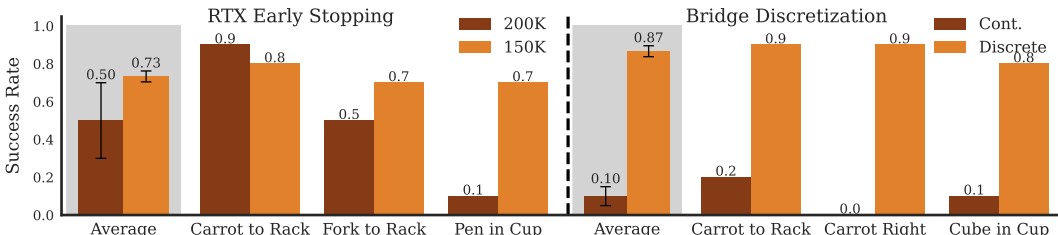

**Figure 4:** Ablations for design choices in Re-Mix. We ablate the effects of **left:** reference model overfitting by selecting a checkpoint once validation loss starts increasing at 150K steps and **right:** using continuous actions for Re-Mix. For ablations, we remove the "Flip Bowl" and 'Cube to Plate" tasks as all Re-Mix variants achieved 100% success.

## A.2    10% Bridge Sub-setting

Here we include results for 10% subsetting of the bridge dataset as described in Section 4.4. In the supplemental material we include videos of rollouts from our experiments.

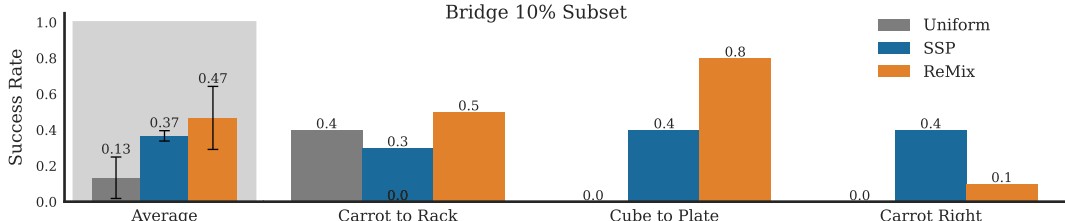

**Figure 5:** Bridge 10% subsetting.

## A.3    Simulation Experiments

We additinally run simulation experiments on the Robomimic NutAssemblySquare task from images [20]. We chose Robomimic because it was collected using human operators like real world datasets. We divided the 300 multi-human demonstrations into six domains by operator, which have "better", "okay", and "worse" labels. We run Re-Mix with the same architecture as described in all other experiments, but train Conditional UNet Diffusion Policies [30] since they performed far better on this benchmark. We evaluate checkpoints for 100 episodes after 400K training steps. The results are included in Table 3 and learned Re-Mix weights are shown in Table 4. We can see that the Re-Mix determined weights outperform uniform weights at both 50% and 25% subsetting. This is likely because Re-Mix up-weights the "better" operators and comparatively down-weights the "worse" ones. Note that the natural or uniform domain weights are not even across all operators. This is because some of the operators take longer to complete the task than others.

| Method | 50% Subsetting | 25% Subsetting |
|---|---|---|
| ReMix | **77/100** | **59/100** |
| Uniform | 53/100 | 39/100 |

**Table 3:** Performance on the RoboMimic NutAssemblySquare task, divided by operator.

| Method | Better 1 | Better 2 | Okay 1 | Okay 2 | Worse 1 | Worse 2 |
|---|---|---|---|---|---|---|
| ReMix | 22.8% | 20.0% | 11.9% | 14.6% | 18.0% | 12.7% |
| Uniform | 9.6% | 13.6% | 18.7% | 14.4% | 20.0% | 23.7% |

**Table 4:** Domain weights used by Re-Mix in comparison to the natural uniform domain weights.

### A.4 Action Distributions

In Fig. 7 we show the action distribution for the BridgeV2 d ataset and in Fig. 8 we show the action distribution for the ToTo dataset, both in log-scale. The BridgeV2 dataset's action distribution is far more normal and symmetric than the ToTo action distribution, which is heavily multi-modal and skew. Robust optimization appears to be more well-behaved on the more normally distributed datasets.

## B   Dataset Details

### B.1   OpenX RTX Subset

We use a subset of the OpenX Embodiment datast similar to that used to train the RT-X models [7]. First, we use the RLDS dataset modification repository (https://github.com/kpertsch/rlds_dataset_mod) used by Octo Model Team et al. [25] to preprocess the raw datasets downloaded from Tensor Flow Datasets [78]. Specifically, we resize all images to $256 \times 256$, and filter the Kuka dataset [76] by an included success key. Note that this does warp images. We use the updated version of the Bridge dataset, available at https://rail.eecs.berkeley.edu/datasets/bridge_release/data/tfds/. The specific composition of the dataset is listed in Table 2. Note that we only train on the primary third-person camera in each dataset. For this reason, we omit the NYU Reacher-grabber dataset [79] which *only* inlcudes wrist cameras. We align all action spaces by converting them to delta cartesian and delta euler angle and binarize all gripper actions.

### B.2   Bridge V2 Dataset

For experiments on bridge-only, we split the bridge dataset into 32 domains. First, we re-downloaded the raw bridge dataset and converted it to RLDS using the DLimp convertor (https://github.com/kvablack/dlimp/). We then partitioned the bridge dataset by domain using the file path metadata field that lists which setting demonstrations were collected in e.g. "toy-kitchen 1" or "toy-sink-3". We then manually group the domains into 32 categories. We omitted data that was collected by a scripted policy, as it did not contain the scene information in the filepath metadata. This means we ended up with around 45,000 training trajectories, instead of the 60K used in the full bridge dataset. In Table 5 we list the natural weights of each of these domains and the learned weights by Re-Mix. We can see that Re-Mix down-weights some of the largest domains and places their weight on smaller domains.

### B.3   Co-Training Datasets.

Below we describe our co-training data and evaluation procedure for the real-world tasks on the WidowX 250 and Franka Panda robots.

**WidowX Tasks**   We evaluate on a 6-DoF WidowX 250 robot on several new pick place tasks in a toy kitchen setting. Our setup is similar to Bridge V2 [4] with a fixed side camera and a blocking controller. Following Walke et al. [4] we use a blocking controller during evaluation. We collect teleoperated demonstrations using an Oculus Quest Headset for motion tracking and co-train on 25 demonstrations for each of the three tasks "Move Cube out of Sink", "Move Cup into Sink", and "Move Fork from Sink to Rack."

During evaluation, we examine generalization on various axes. The "Carrot to Rack" task tests generalization to picking up a new type of target object, "Fork to Rack" tests new unseen object positions, "OOD Cup" tests an object with different shape, "Cube to Plate" and "Cube to Cup" test generalization to new containers, and "Carrot to Right" tests generalization to both a new target object and a new motion. For each of these tasks, we first take a goal image and then evaluate our policies with fixed object locations for up to 100 seconds, stopping early if the robot or objects reach unrecoverable states. For "Carrot to Rack" we do five trials with the carrot facing down and five trials with it facing upwards. For "Fork to Rack" we use an unseen initial position to the right side of the sink and rotate the fork left 45 degrees for five episodes and to the right 45 degrees for the other five. We also tested on an additional "Cube Distractor" task in which the robot has to move the cube to the sink with a distractor object present. We ommited this task

| Domain | Uniform Weight | ReMix Weight |
|---|---|---|
| 0 toykitchen2 | 0.18728751 | 0.0961817 |
| 1 datacol2_tabletop_dark_wood | 0.094527 | 0.04846529 |
| 2 toykitchen1 | 0.069307 | 0.07683 |
| 3 toykitchen6 | 0.06940527 | 0.0573625 |
| 4 datacol2_toykitchen7 | 0.07133783 | 0.06905 |
| 5 datacol2_toykitchen2 | 0.0432927 | 0.03651583 |
| 6 toykitchen7 | 0.032803 | 0.03538789 |
| 7datacol2_folding_table | 0.038522 | 0.0809778049 |
| 8 datacol1_toykitchen6 | 0.03606622 | 0.037404168 |
| 9 datacol2_robot_desk | 0.025810027 | 0.034152 |
| 10 datacol2_toykitchen6 | 0.02394393 | 0.02740302 |
| 11 deepthought_folding_table | 0.0272809 | 0.013906823 |
| 12 datacol2_laundry_machine laundry_machine | 0.02582954 | 0.0396389 |
| 13 datacol2_toykitchen5, toykitchen5 | 0.0337366 | 0.049943 |
| 14 deepthought_toykitchen2 | 0.0253313 | 0.013434348 |
| 15 deepthought_robot_desk | 0.01978364 | 0.032410502 |
| 16 tabletop_dark_wood | 0.0219985 | 0.024691 |
| 17 datacol2_toysink2 toysink2_bww | 0.0225748 | 0.0198516 |
| 18 toykitchen2_room8052 | 0.01083554 | 0.0295857 |
| 19 deepthought_toykitchen1, datacol1_toykitchen1 | 0.01868 | 0.04047 |
| 20 datacol2_foldtable_tray, minsky_foldtable_tray, datacol2_toykitchen7_tray | 0.037856699 | 0.0484 |
| 21 toysink3_bww, toysink3 | 0.01235829 | 0.014877 |
| 22 datacol2_toykitchen1 | 0.01155453 | 0.02194 |
| 23 toysink1_room8052 toysink1 | 0.00979455 | 0.01831014 |
| 24 tool_chest | 0.00471524 | 0.00878 |
| 25 toysink5 | 0.00405418 | 2.78E-05 |
| 26 whiteboard | 0.006774 | 0.0129337 |
| 27 toykitchen4 | 0.00371938 | 0.00537445 |
| 28 toysink4 | 0.00289793 | 1.80E-05 |
| 29 toykitchen3 | 0.00124406 | 2.72E-05 |
| 30 realkitchen1_dishwasher | 0.00202648 | 0.000541 |
| 31 tabletop_light_wood, tabletop_white, realkitchen1_counter | 0.004647549 | 0.005079152 |

**Table 5:** Learned weights by Re-Mix on the Bridge V2 dataset.

from Fig. 2 as no baselines were able to complete the task, which heavily skewed performance. We include results for this task in Appendix B.3.

| Method | ReMix | Human | Uniform |
|---|---|---|---|
| Success Rate | 7/10 | 0/10 | 0/10 |

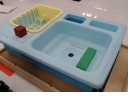

**Figure 6:** Performance on the cube Distractor task. Right: Depiction of the Cube Distractor task.

## B.4 Franka Tasks

We evaluate on a Franka Panda robot on several pick place tasks on a tabletop. We use a fixed over the shoulder camera We co-train on 25 teleoperated demonstrations for each of the tasks "Pen into Cup," where we put a pen into a cup from 5 different start locations, and "Flip Bowl," where a bowl is flipped into a drying rack. For the "Pen into Cup" task we use a different pen than in co-training. However, because our franka embodiment with the Robotiq 2F-85 is not found in our pre-training datasets, we evaluate the same tasks as we co-trained on. We evaluate each start location of the pen twice from a new set of predifined positions. As in the WidowX evaluations, we take a goal image for each task and evaluate for up to 100 seconds using a 10Hz controller without blocking control.

|                  | RTX              | Bridge           |
| ---------------- | ---------------- | ---------------- |
| Batch Size       | 512              | 384              |
| Action Chunk     | 4                | 2                |
| Image Resolution | $224 \times 224$ | $224 \times 288$ |

**Table 6:** Hyperparameters

## C Training Details

**Architecture.** We borrow our architecture from [4] with a few minor changes. Our policies takes as input a history of two consecutive frames and a single goal image and output a sequence of actions via DDPM [80].

First, we preprocess all images to fit between -1 and 1. Then, we channel-wise concatenate both the goal image and a grid containing the position of each pixel in $(x,y)$ space also normalized between -1 and 1. Images are then fed to a ResNet 50 encoder, which employs global average pooling on the output to obtain a 512 dimension representation for each image. Both image representations are then concatenated and fed to a diffusion action prediction head.

**Hyperparameters.** We use a cosine decay learning rate schedule with an initial learning rate of 0.0002. We train all models for 400K steps and evaluate the final checkpoint, except for Bridge 10% subsetting, which we found to perform better after 200K steps. More detailed hyperparameters are found in Table 6. Note that there are some differences between bridge and RTX which were made for computational reasons – we iterated faster on the bridge dataset before scaling to RTX. We also did maintained aspect ratio for bridge, hence the different image input size, but did not for RTX follow Octo Model Team et al. [25]. We apply data augmentation to all images consistently across the time horizon and goal image (meaning that the goal image and all past images of each example have the same augmentation applied). We use random resize cropping, brightness, contrast, and hue randomization. For k-means in SSP for Bridge we set $k=32$, equal to the number of domains used for Re-Mix.

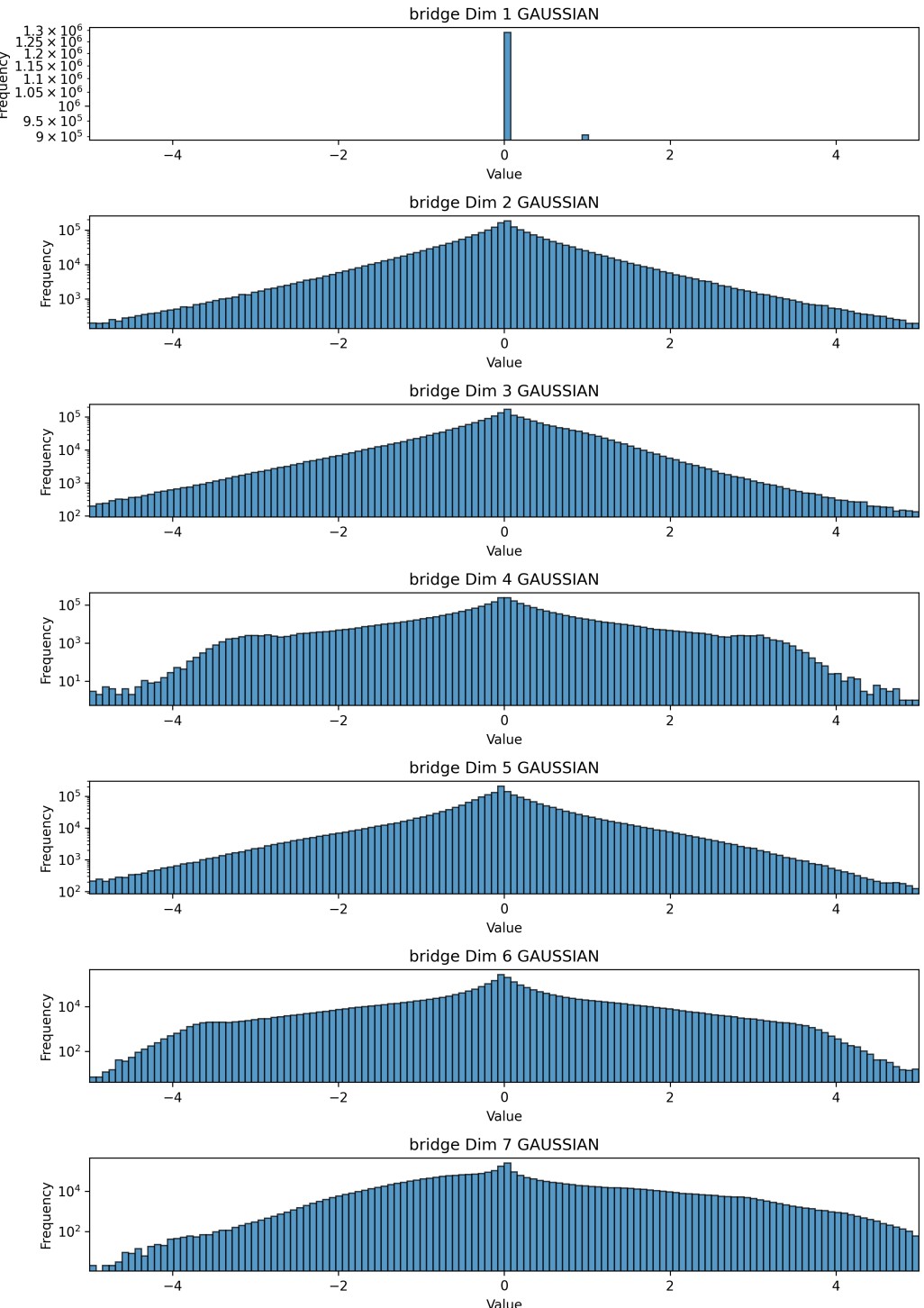

**Figure 7:** Action distributions for Bridge.

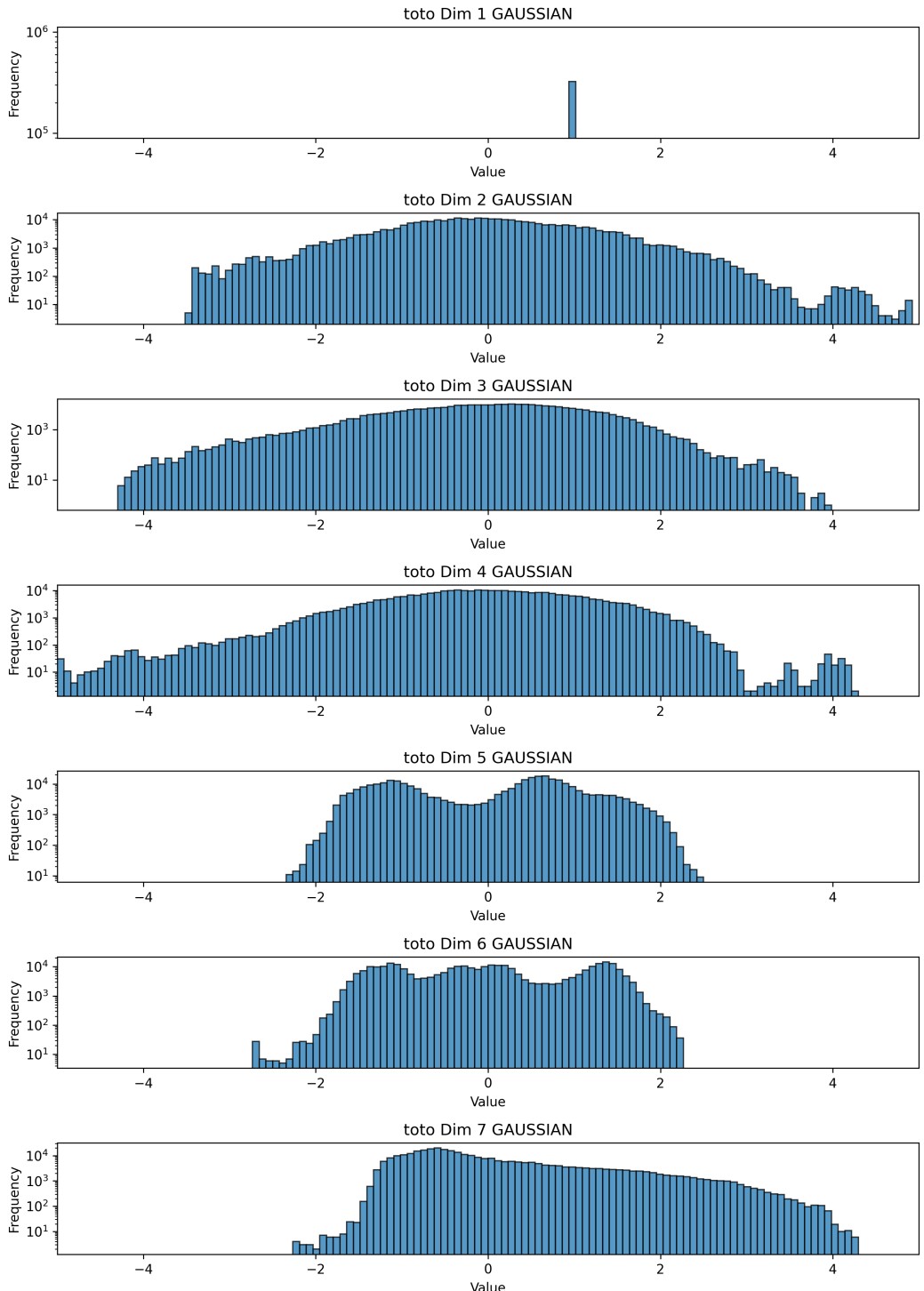

**Figure 8:** Action distributions for Toto.

