# OpenReview forum: "ReMix: Optimizing Data Mixtures for Large Scale Imitation Learning"
_robot-learning.org/CoRL/2024/Conference — CoRL 2024_

### Official Review · Reviewer_zvQ8 · 2024-07-04
**Rebranding of DoReMi for LLMs in Robotics**

**Originality:** 1
**Technical Quality:** 1
**Clarity Of Presentation:** 4
**Potential Impact:** 2
**Recommendation:** 1
**Confidence:** 4

**Review:**

This paper applies DoReMi [1] to the field of robotics dataset for data mixture. The formulation is largely the same: using minimax optimization, applying GroupDRO and using a reference model.

1. The name of the excess loss is from [1].
2. The minimax optimization problem is the same as [1].
3. The GroupDRO method is from [1].

The only differences are:

1. changing the loss function from likelihood to policy prediction.
2. switching the dataset to robotics dataset like OpenX.

Also, it conducts analysis on how Re-Mix weights impact model performance, how well Re-Mix subset datasets do, and the proportion of each subset in RT-X.

Overall I think the significance of this paper is limited compared to DoReMi as most formulation is the same. I'm not sure the acceptance standard of CoRL but simply changing the application shouldn't be considered a novel paper.

[1] Xie, Sang Michael, et al. "Doremi: Optimizing data mixtures speeds up language model pretraining." Advances in Neural Information Processing Systems 36 (2024).

**Quality Of The Limitations Section:**

3

**Questions For Rebuttal:**

I don't have specific questions in mind.

**Robotics Focus:**

4

**Summary Of Paper:**

This paper applies DoReMi into the field of robotics dataset for data mixing.

**Summary Of Recommendation:**

Overall the originality is quite limited. I would not recommend this paper for acceptance.

---

### Official Review · Reviewer_Q6JL · 2024-07-14
**Effective method to optimize cross-embodiment datasets for foundation models for better generalization across datasets.**

**Originality:** 3
**Technical Quality:** 4
**Clarity Of Presentation:** 4
**Potential Impact:** 3
**Recommendation:** 4
**Confidence:** 4

**Review:**

Strengths
- the paper tackles an important problem in the context of foundation policies to find the optimal data mix for the best generalization
- the method is the first one in the context of robotics to curate large-scale datasets for imitation learning. A novel but non explored problem in the context of foundation models for robotics
- The results are evaluated thoroughly and the substantial performance improvement underline the effectiveness across several real robot tasks
- Clever usage of discrete action binning to migate the problem of L2 BC loss functions
- the main section of the paper does a good job  of explaining the problem of data mixing and why standard methods fail. Given this paper tackles a novel problem, it helps to understand the problem of data mixing and how Re-mix solves it.
- the results of the curated smaller dataset do outperform the full tested not remixed dataset, which enables more efficient and faster training


Weaknesses
- the proposed method is very expensive and requires training twice on large-scale datasets
- the method is an adaption of prior work from NLP for the robotics domain

**Quality Of The Limitations Section:**

3

**Questions For Rebuttal:**

- while I appreciate the various experiments in the real world. Could the authors try to add 1 or 2 sim environments to the dataset mixture and report results in this sims? This would allow better benchmarking of future work in this field by using available simulation.
- given stronger BC policies that have been introduced recently such as VQ-BeT would it be possible to reduce the double training when with a more expressive policy?

—

The authors were able to resolve my concerns and added my requested experiments for better reproduction using simulation. This improves the paper and I maintain my high score.

**Robotics Focus:**

4

**Summary Of Paper:**

The paper introduces Re-Mix, a robust optimization method tailored to find the composition of large-scale imitation learning datasets that consist of different robot datasets. The method takes ideas used in NLP to find optimal dataset mixtures and adapts them for robotic datasets by solving several challenges related to robotic data. The experiments show that policies trained with Re-Mix outperform policies on default robot dataset mixes and mixes by robot experts. Further, by using the generated importance weights the authors show that policies trained on better datasets can reduce the dataset by 75% and still outperform default mixes. while I appreciate the various experiments in the real world. Could the authors try to add 1 or 2 sim environments to the dataset mixture and report results in this sims? This would allow better benchmarking of future work in this field by using available simulation.

**Summary Of Recommendation:**

The paper tackles an important and novel problem of optimizing the dataset of foundation policies and contributes relevant contributions for the Robotics community. Overall, the work is timely and I am excited for the possibilities, especially with the improved efficiency in training with smaller datasets optimized by re-mix. In addition, several experiments in the real world are conducted to verify various claims. Thus, I recommend accepting this paper.

---

### Official Review · Reviewer_aygU · 2024-07-20
**Review for Submission634**

**Originality:** 4
**Technical Quality:** 5
**Clarity Of Presentation:** 5
**Potential Impact:** 4
**Recommendation:** 4
**Confidence:** 5

**Review:**

- Quality : Addresses an important problem in large scale robot imitation learning. The hypotheses and supportive experiments are well thought out. Provides a good comparison and clear delineation of why data curation techniques from vision and language domains cannot be directly transferred to robotics.
- Clarity : Exceptionally well written, making it a pleasure to read.
- Originality and significance : As the field moves towards large scale imitation learning for robotics, this paper offers a more principled solution than handcrafted data mixtures previously used. Highly relevant and original contribution.

**Strengths**

- Offers clear explanations and rationale for the modifications needed to adapt vision and language techniques to robotics.
Provides a principled approach to optimizing training data mixtures, which is important for effective large-scale imitation learning in robotics.
- Extensive analysis for each domain has also been provided in the appendix with comprehensive experiment details.

**Weaknesses**

- It would have been interesting to see more diverse tasks being picked for evaluation. For example - carrot to rack and fork to rack tasks potentially involve very similar action distributions.

**Quality Of The Limitations Section:**

3

**Questions For Rebuttal:**

It would be interesting to see if more diverse tasks can be added to the evaluations. Intuitively, carrot to rack and fork to rack tasks potentially involve very similar action distributions. If not, adding plots of action distribution similarity could also clarify this.

**Robotics Focus:**

4

**Summary Of Paper:**

Re-Mix is a min-max approach for optimizing training data mixtures in large-scale robot imitation learning. Using group distributionally robust optimization, Re-Mix dynamically adjusts domain weights within robotic datasets to maximize performance across various domains (i.e., different robot datasets). As opposed to current practices that rely on expert intuition for data curation, Re-Mix offers a systematic method to assess domain significance through a min-max optimization framework. This approach results in better performance compared to both uniform and expert-curated data mixtures.

**Summary Of Recommendation:**

This paper presents a principled approach to selecting data to scale up imitation learning from large robot datasets. It provides clear intuitions and is well written. This is particularly relevant to the current paradigm in robotics and could be widely applicable.

---

### Author Rebuttal · Authors · 2024-08-07

We would like to thank the reviewers and meta-reviewer for taking the time to review our work. We are pleased to hear that it “addresses an important problem” (Reviewer aygU) and has a “substantial performance improvement” (Reviewer Q6JL). Here we provide a general response, and address reviewer specific concerns below their individual reviews.

**Additional evaluation tasks**

We have run additional evaluations of the RTX ReMix policy and baselines on three additional tasks for Reviewer Q6JL and the MetaReviewer to increase the evaluation diversity:

* “OOD Cup”: The robot has to move an out-of-distribution cup object  to the sink, in a different direction than the carrot and fork tasks.
* “Cube To Plate”: The robot moves a cube from the sink to the top of an unseen plate with a different goal location than the carrot and fork.
* “ Cube Distractor”: The robot moves the cube from a different initial location out of the sink with a large green distractor block.

A revised version of Figure 1 depicting the tasks and results can be found in the updated manuscript.

| Method  | Average | OOD CupToSink | CubeDistractor | CubeToPlate | CarrotToRack | ForkToRack | FrankaPen | FrankaBowl |
|---------|---------|---------------|----------------|-------------|--------------|------------|-----------|------------|
| ReMix   | **0.7857**  | **0.7**           | **0.7**            | 0.8         | **1.0**          | **0.8**        | 0.6       | **0.9**        |
| Human   | 0.414   | 0.0           | 0.0            | 0.8         | 0.7          | 0.4        | 0.6       | 0.4        |
| Uniform | 0.357   | 0.1           | 0.0            | **0.9**         | 0.3          | 0.3        | 0.6       | 0.3        |

We find that the ReMix model performs better in the presence of distractors and at dealing with the OOD cup.

**Simulation Results**

In response to reviewer Q6JL we have run simulation experiments on the Robomimic NutAssemblySquare task from images. We divide the 300 demo multi-human dataset into six domains by operator. We train ReMix with the architecture from Section 4, then train Diffusion Policy (Chi et al. 2023) on subsets of the data according to the ReMix and Uniform weights. We run 100 eval episodes after 400K training steps.

| Method  | 50% Subset | 25% Subset |
|---------|------------|------------|
| ReMix   | 77/100     | 53/100 |
| Uniform | 59/100     | 34/100 |

We have additionally included these results, and the learned ReMix weights in Appendix A.2.

---

### Decision · Program_Chairs · 2024-09-04

**Decision:**

Accept

**Comment:**

**Paper summary**

This paper adapts techniques from NLP to improve the generalizability of robot foundation models by re-weighting datapoints in large-scale datasets. The paper presents extensive experiments in real-world and simulated tasks, demonstrating how the method improves the data-efficiency and generalizability of the trained model compared to the original, uniform data weighting.

**Review summary**

Summary of strengths:
+ The paper addresses a key problem in robot learning: optimizing training data mixtures. The paper has the potential for high impact in the CoRL community.
+ The evaluation is thorough, involves real-world experiments, and demonstrates how the remixed dataset enables faster training and higher-performance in the trained model.
+ The paper is well-written and clear.

Summary of weaknesses:
- The proposed method borrows heavily from prior work (DoReMi). While the paper describes how the application is different (robot policies vs language models), it is unclear whether the proposed method itself is sufficiently novel. **[Edit: the rebuttal argues that the paper overcomes several practical barriers that otherwise prevent DoReMi from being simply reused in robotics.]**
- The evaluation could be improved with added variance in the environments and tasks. **[Edit: the rebuttal includes additional experiments in higher-variance tasks.]**

**Response to rebuttal**

Following the rebuttal, the reviewers remain split about the originality of the paper's contribution; 1 reviewer advocates for a strong reject (on the basis of the paper being a "rebrand" of DoReMi for a robotics application) and 2 reviewers advocate for a strong accept (on the basis of the paper overcoming practical barriers to adapt DoReMi for robotics and thoroughly demonstrating its effectiveness for the robot learning community).

I lean toward the latter view. This paper provides compelling evidence to demonstrate the importance and practical effect of data mixtures for robot learning. I believe this will be a highly useful and practical contribution for the CoRL community.


**Recommendations for improvement**

The revised paper should include the rebuttal's argument for why the robotics domain requires unique considerations compared to language modeling with DoReMi.